# Detailed Changes in Oxygenation following Awake Prone Positioning for Non-Intubated Patients with COVID-19 and Hypoxemic Respiratory Failure—A Historical Cohort Study

**DOI:** 10.3390/healthcare10061006

**Published:** 2022-05-29

**Authors:** Tomotaka Koike, Nobuaki Hamazaki, Masayuki Kuroiwa, Kentaro Kamiya, Tomohisa Otsuka, Kosuke Sugimura, Yoshiyuki Nishizawa, Mayuko Sakai, Kazumasa Miida, Atsuhiko Matsunaga, Masayasu Arai

**Affiliations:** 1Department of Intensive Care Center, Kitasato University Hospital, Sagamihara 252-0329, Japan; tkoike@kitasato-u.ac.jp; 2Department of Rehabilitation, Kitasato University Hospital, Sagamihara 252-0329, Japan; hamanobu0317@gmail.com (N.H.); miidak@kitasato-u.ac.jp (K.M.); 3Department of Anesthesiology, School of Medicine, Kitasato University, Sagamihara 252-0373, Japan; masa9618@yahoo.co.jp (M.K.); o-chan@kitasato-u.ac.jp (T.O.); jtaro1031@yahoo.co.jp (K.S.); yoshiyukinishizawa@yahoo.co.jp (Y.N.); 3mh6ee@bma.biglobe.ne.jp (M.S.); 4Department of Rehabilitation, School of Allied Health Sciences, Kitasato University, Sagamihara 252-0373, Japan; atsuhiko@kitasato-u.ac.jp; 5Division of Intensive Care Medicine, Department of Research and Development Center for New Medical Frontiers, School of Medicine, Kitasato University, Sagamihara 252-0373, Japan; hiroma557@yahoo.co.jp

**Keywords:** awake prone positioning, COVID-19, hypoxemic respiratory failure, oxygenation, acute respiratory distress syndrome (ARDS), chest physiotherapy, ROX index

## Abstract

Few studies have reported on the effectiveness of awake prone therapy in the clinical course of coronavirus disease (COVID-19) patients. This study aimed to investigate the effects of awake prone therapy during spontaneous breathing on the improvement of oxygenation over 3 weeks for COVID-19 acute respiratory failure. Data of consecutive COVID-19 patients with lung disorder with a fraction of inspired oxygen (F_I_O_2_) ≥ 0.4 and without tracheal intubation were analyzed. We examined changes in SpO_2_/F_I_O_2_, ROX index ((SpO_2_/F_I_O_2_)/respiratory rate) and the seven-category ordinal scale after the initiation of F_I_O_2_ ≥ 0.4 and compared these changes between patients who did and did not receive prone therapy. Of 58 patients, 27 received awake prone therapy, while 31 did not. Trend relationships between time course and change in SpO_2_/F_I_O_2_ and ROX index were observed in both groups, although a significant interaction in the relationship was noted between prone therapy and change in SpO_2_/F_I_O_2_ and ROX index. The seven-category ordinal scale also revealed a trend relationship with time course in the prone therapy group. The awake prone therapy was significantly associated with a lower rate of tracheal intubation. In patients with COVID-19 pneumonia treated with F_I_O_2_ ≥ 0.4, awake prone therapy may improve oxygenation within two weeks.

## 1. Introduction

The coronavirus disease (COVID-19) pandemic has caused a global health threat. Although most patients with COVID-19 have asymptomatic or mild illnesses, such as cough or fever, approximately 14% develop severe illnesses, including hypoxic respiratory failure and/or acute respiratory distress syndrome (ARDS), leading to increased mortality [1]. Patients with severe respiratory failure due to COVID-19 may also experience viral pneumonia-induced ARDS [2]. As the established treatment for respiratory failure due to COVID-19 has been limited in the clinical setting, many health practitioners are facing the problem of restricted intensive care implementation. 

Prone position therapy has been used as a strategy to improve oxygenation by ameliorating the distribution of ventilation and blood flow in patients with ARDS receiving invasive positive pressure ventilation [3], including for viral pneumonia-induced ARDS [4,5,6]. Recently, several studies have reported the efficacy of prone therapy in the awake state on clinical outcomes in patients with COVID-19 [7,8,9,10,11,12]. A rapid review has also revealed an immediate improvement in oxygenation with prone positioning in these patients [13]. Additionally, there have been no reports of major adverse events related to the prone position therapy, and it has been reported to be safe, regardless of comfortability, under awake conditions [10,14]. Conversely, the largest prospective cohort study to date has reported that awake prone position under high-flow oxygen therapy for very severe patients cannot prevent tracheal intubation, although it may increase the risk of delayed intubation [9]. However, there are few reports on the effectiveness of awake prone therapy in the detailed clinical course of patients with COVID-19. Here, we conducted a historical cohort study to clarify the effect of prone positioning in patients with COVID-19 pneumonia with implementation under spontaneous breathing and awake conditions. This study aimed to investigate the effects of awake prone positioning on longitudinal changes in oxygenation and the avoidance of tracheal intubation and mechanical ventilation in patients with COVID-19 pneumonia and respiratory failure.

## 2. Materials and Methods

### 2.1. Study Design Sampling

This study was a before–after historical cohort study. We retrospectively reviewed consecutive patients with COVID-19 aged ≥ 20 years who were diagnosed with severe acute respiratory syndrome coronavirus 2 (SARS-CoV-2) infection by a positive polymerase chain reaction test and admitted to our hospital between 1 October 2020, and 31 March 2021. Eligible patients had hypoxemic respiratory failure with a fraction of inspired oxygen (F_I_O_2_) ≥ 0.4. The study protocol was approved by the Ethics Committee of our institution (B20-370) and followed the Strengthening Reporting of Observational Studies in Epidemiology reporting guidelines [15] for cohort studies. The following patients were excluded from this study: those who were pregnant, patients already intubated at admission to our institution or those who were immediately intubated, and patients who received invasive positive pressure ventilation. The historical cohorts consisted of pre-intervention phase (admitted to the ICU from 1 October 2020 to 1 December 2020) and the intervention phase (from 1 December 2020 to 31 March 2021). We initiated the awake prone position therapy for respiratory failure in patients with COVID-19 and F_I_O_2_ ≥ 0.4 and compared the changes in oxygenation, ROX index, and the clinical course between pre- and post-initiation of the prone therapy.

### 2.2. Protocol of Awake Prone Therapy

Prone positioning was initiated when the F_I_O_2_ reached ≥ 0.4, starting in December 2020. If the patient was conscious, the respiratory therapy team provided a full explanation of the procedure and its expected efficacy and safety to the patient, and prone therapy was initiated if the patient agreed with these. The respiratory therapy team stayed for at least 30 min at the prone therapy initiation to monitor the patient’s comfort and safety. Prone therapy was discontinued if the patient developed intolerable respiratory distress, tachypnea >35 bpm, or new unacceptable back pain during prone therapy. If prone therapy was difficult to continue due to pain, even without the abovementioned symptoms, it was discontinued. If the patient could maintain the prone position for >30 min in the first session, we considered the patient to be eligible for prone therapy and performed it at least twice a day. Depending on their ability in performing the prone position independently, patients were allowed to assume the prone position without special assistance. When the patient was unable to perform the prone position independently, the medical staff provided dynamic assistance to minimize the oxygen consumption associated with the prone position. The time and number of days of the prone therapy were recorded. Awake prone positioning continued until the patient was discharged. The awake prone positioning continued even when the patient’s respiratory status deteriorated and the oxygen delivery interface was changed (e.g., simple oxygen mask to HFT). The awake prone positioning was discontinued when the patient was intubated; however, the data for event of intubation, course of F_I_O_2_, and seven category ordinal scale were collected.

### 2.3. Oxygen Weaning and Pain Management

The respiratory therapy team, intensivists, and emergency physicians assessed respiratory and systemic status if F_I_O_2_ ≥ 0.4 was reached. Fifty-one cases (88%) were examined for BGA when they reached F_I_O_2_ ≥ 0.4. The initial setting of oxygen flow rate was maintained to reach SpO_2_ 92% to 94% with reference to blood gas analysis.

Patients with symptoms of respiratory distress and/or a respiratory rate of ≥26 breaths were, in principle, placed on high-flow nasal cannula (HFNC). HFNC was also applied if there was a rapid progression of SpO_2_ decrease within 1 day. Tracheal intubation was performed in cases of hypoxemia that could not be overlooked even if the HFNC setting was increased to the maximum, and in cases where tachypnea and respiratory distress did not improve. Non-invasive positive pressure ventilation (NPPV was not used in principle, although it was applied only when the physician did not consider other indications in the do-not-intubate (DNI) order. Dexmedetomidine and opioids were used as rescue therapy or palliative care if respiratory distress remained severe under HFNC. In patients with the DNI order, opioids were used to relieve respiratory distress if the patient complained of respiratory distress despite increased O_2_ flow or changed HFNC settings. SpO_2_ was continuously monitored under oxygen inhalation and recorded 6 times a day (every 4 h), except when desaturation occurred as noted. The SpO_2_ recorded at 9:00–10:00 AM was adopted as the representative SpO_2_ of the day in the data. F_I_O_2_ of HFNC, NPPV, and O_2_ flow rate of the simple oxygen mask was reduced by 0.1 (1 l/min for the simple oxygen mask) if SpO_2_ was ≥96%. As a rule, oxygen reduction attempts were made once a day.

### 2.4. Clinical Characteristics and Treatment Information 

Clinical characteristics such as age, sex, body mass index, number of days from disease onset to F_I_O_2_ of 0.4, medical history, smoking history, blood examinations on admission, and maximum values (serum ferritin, fibrinogen-fibrin degradation product, D-dimer, lactic acid dehydrogenase, and C-reactive protein), antiviral medication (favipiravir or remdesivir), steroid use, the highest F_I_O_2_, and intubation order were obtained from medical records. Data on respiratory rate and symptoms (dry cough, fever >38 °C, confusion or disorientation, tachypnea >25, dyspnea) at the initiation of F_I_O_2_ ≥ 0.4 were also collected.

### 2.5. Assessment of Oxygenation and Clinical Outcomes on Awake Prone Therapy 

The SpO_2_/F_I_O_2_ at 3 days, 1 week, 2 weeks, and 3 weeks after the initiation of F_I_O_2_ ≥ 0.4 were assessed as outcomes of changes in oxygenation. Since most of the non-intubated patients were monitored noninvasively, arterial blood gas analysis was not routinely evaluated. Therefore, SpO_2_/F_I_O_2_ values, reportedly correlated with the PaO_2_/F_I_O_2_ ratio [16], was used to assess an oxygenation. The ROX index [17] ((SpO_2_/F_I_O_2_)/respiratory rate), as an index to predict the need for MV in patients with acute respiratory failure [18,19], was also collected at the same time point of SpO_2_/F_I_O_2_ for assessing ventilatory efficiency and need for mechanical ventilation. The F_I_O_2_ was defined as follows: nasal cannula of 1 L/min as 0.24, 2 L/min as 0.28, 3 L/min as 0.32, and 4 L/min as 0.36; simple oxygen mask of 5 L/min as 0.4 and 6 L/min as 0.5; non-rebreather mask of 6 L/min as 0.6, 7 L/min. as 0.7, 8 L/min. as 0.8, 9 L/min. as 0.9, and 10 L/min as 1.0. Scores on the seven-category ordinal scale [16], which consisted of the following categories: 1, not admitted to hospital with resumption of normal activities; 2, not admitted to hospital, but unable to resume normal activities; 3, admitted to hospital but not requiring supplemental oxygen; 4, admitted to hospital but requiring supplemental oxygen; 5, admitted to hospital requiring HFNC, non-invasive mechanical ventilation, or both; 6, admitted to hospital requiring extracorporeal membrane oxygenation, invasive mechanical ventilation, or both; and 7, death, were assessed at 1, 2, 3, and 4 weeks after the initiation of F_I_O_2_ ≥ 0.4 as clinical outcomes. Tracheal intubation and hospital discharge (discharge to home with or without oxygen therapy, to long-term care hospital with or without oxygen, and death) were also recorded. In patients receiving prone therapy, the number of days from onset and from F_I_O_2_ ≥ 0.4 to the start of prone therapy, the number of prone sessions per day, the number of oxygen devices during prone therapy, and the impressions of the prone position were recorded. Endpoints were set from the time F_I_O_2_ reached 0.4 until discharge, or 4 weeks later. In patients who could not perform the prone position within the term of prone therapy, we reviewed their symptoms and endpoints from the medical records.

### 2.6. Statistical Analysis 

Continuous variables are expressed as median and interquartile range, and categorical variables are expressed as numbers (percentages). We compared the clinical characteristics, treatment information, condition at hospital discharge, and clinical outcomes including SpO_2_/F_I_O_2_, ROX index, and the seven-category ordinal scale at each phase between prone therapy and non-prone therapy using the Mann–Whitney U test and the Chi-square test, as appropriate. The trend relationships of time-course with changes in F_I_O_2_ and the seven-category ordinal scale were analyzed using the Jonckheere–Terpstra test and Cochran–Armitage test, respectively, and the statistical interactions of prone therapy with changes in F_I_O_2_ and seven-category ordinal scale. Data analysis was performed using the SPSS statistical software (SPSS 21.0: SPSS; Chicago, IL, USA) and R version 3.1.2 (R Foundation for Statistical Computing, Vienna, Austria), and a two-tailed *p* value of <0.05 was considered significant.

## 3. Results

Figure 1 shows the patient flowchart. In the potential population of 291 patients admitted to our hospital due to COVID-19 infection during the study period, 63 patients had F_I_O_2_ ≥ 0.4. Of the 63 patients, 31 were classified in the non-prone group, and 32 were classified in the prone group based on the before–after historical cohort. Of the 32 patients in the prone group, five patients were also excluded from the analysis for discontinuing prone therapy and were therefore counted separately as cases in which the continuation of the prone position could not be established. Consequently, both groups, with 31 patients in the non-prone group and 27 patients in the prone group, were compared.

Baseline F_I_O_2_ in all patients had a median of 0.60 interquartile range of 0.5–0.8. Table 1 presents the baseline characteristics of the patients. No statistical differences were observed between both groups in terms of respiratory conditions, comorbidities, smoking history, variables of blood examination, symptoms, rescue therapy, palliative care, and medical treatment, except for a higher rate of remdesivir prescription in the prone group than in the non-prone group (Table 2).

Figure 2a shows the SpO_2_/F_I_O_2_ after initiation of F_I_O_2_ ≥ 0.4 between both groups. SpO_2_/F_I_O_2_ tended to increase in both groups during the 3-week observation period after the initiation of F_I_O_2_ ≥ 0.4 (*p* < 0.001, respectively). Conversely, SpO_2_/F_I_O_2_ at 1 and 2 weeks after initiation of F_I_O_2_ ≥ 0.4 was significantly higher in the prone group than in the non-prone group, and a significant interaction in the association between prone therapy and change in SpO_2_/F_I_O_2_ (*p* = 0.016) was noted. SpO_2_/F_I_O_2_ at 1 week after initiation of F_I_O_2_ ≥ 0.4 was significantly higher in the prone group than in the non-prone group (*p* = 0.03). Figure 2b shows the ROX index after initiation of F_I_O_2_ ≥ 0.4 between both groups. The ROX index also tended to increase in both groups during the 3-week observation period after the initiation of F_I_O_2_ ≥ 0.4 (*p* < 0.001, respectively). The ROX index at 1 and 2 weeks after initiation of F_I_O_2_ ≥ 0.4 was significantly higher in the prone group than in the non-prone group, and a significant interaction in the association between prone therapy and change in SpO_2_/F_I_O_2_ (*p* = 0.014) was noted. Figure 3 shows the seven-category ordinal scale between the groups after initiation of F_I_O_2_ ≥ 0.4 between the groups. The seven-category ordinal scale in the prone group indicated a significant trend relationship with time course (*p* = 0.002), although this association was not observed in the non-prone group (*p* = 0.102). Additionally, the values of the seven-category ordinal scale at 1 and 4 weeks after initiation of F_I_O_2_ ≥ 0.4 were significantly lower in the prone group than in the non-prone group (*p* = 0.03 and *p* = 0.02, respectively). No statistical interaction in the association between prone therapy and changes in the seven-category ordinal scale (*p* = 0.89) was observed. 

No significant difference in the mortality rate or the percentage of patients requiring continued oxygenation at discharge or transfer was noted. Information on the other clinical outcomes between the two groups is shown in Table 3. Prone therapy was significantly associated with a lower rate of tracheal intubation, although no statistical differences in the condition at hospital discharge were observed between both groups.

The time course after initiation of F_I_O_2_ ≥ 0.4 is shown in Table 2. The prone therapy was started as soon as F_I_O_2_ ≥ 0.4, but the median F_I_O_2_ just before the start of prone position was 0.6 because oxygenation was ensured according to the adjustment of oxygen therapy described above. The number of prone positions performed per day was median 2 interquartile range 2–3, the median prone position time was median 180 min interquartile range 120–240, and the number of practice days of prone therapy was median 13 days interquartile range 7–16. Of these cases, 14 (52%) had an antitussive effect, and 10 (37%) had improved dyspnea.

In the five excluded patients who could not sustain prone positioning, the main causes were tachypnea with a respiratory rate >25 bpm (80%) and dyspnea (60%) during prone positioning. Of these five patients, two patients died (Table 4).

## 4. Discussion

Our study revealed that prone positioning for non-intubated patients with COVID-19 and respiratory failure with F_I_O_2_ ≥ 0.4 was associated with the improvement of short-term SpO_2_/F_I_O_2_ reduction and ROX index. Recovery of the seven-category ordinal scale in the prone group also indicated a significant association with the time trend, while no such trend was observed in the non-prone group. 

To the best of our knowledge, there are few reports investigating the trend of oxygenation and patient status for 3 weeks after starting prone therapy with an F_I_O_2_ of 0.4. Several studies have reported that the prone position improves oxygenation in non-intubated patients with COVID-19 respiratory failure [5,10,11,14], which is consistent with our findings. Conversely, Fernando et al. [9] have reported that prone positioning did not reduce the risk of intubation, but rather delayed the risk of intubation. However, the patients in their study had an oxygen rate of 15 L/min in a non-rebreather simple oxygen mask with SpO_2_ ≤ 93%, which is much more severe than our criteria of F_I_O_2_ ≥ 0.4 for prone induction. Moreover, reports that started prone therapy at an oxygenation capacity similar to that of the patients in our study have similarly reported improvement in oxygenation [10,20]. Therefore, for patients with respiratory failure, natural airway, and COVID-19, we considered the awake prone intervention as more effective on oxygenation, while the deterioration of oxygenation was relatively mild, especially when the F_I_O_2_ was 0.4. 

Here, the prone group tended to have a faster withdrawal from high-concentration oxygen by improvement of oxygenation, lower rate of tracheal intubation, and better clinical endpoints in the seven-category ordinal scale despite no significant differences in baseline characteristics, including the severity of COVID-19. Prolonged high-concentration oxygenation is also known to cause histological changes similar to ARDS [21]. Therefore, early withdrawal from high-concentration oxygen by prone therapy may reduce lung damage due to oxygen toxicity. Our results revealed that SpO_2_/F_I_O_2_ was significantly higher in the prone group at 1 week, with a median of 0.30. Although the introduction of the prone position tended to be delayed by approximately 1 d due to various medical problems, and there were many cases in which F_I_O_2_ increased during that time, resulting in an F_I_O_2_ of 0.6, the introduction of the prone position was able to lower F_I_O_2_ early, suggesting that the prone position contributes significantly to the reduction of oxygen requirements. In patients with COVID-19 respiratory failure and managed with HFNC, ROX index >3.0 at 2, 6, and 12 h after HFNC initiation was reported to be highly sensitive in identifying successful HFNC [19]. However, ROX index before the initiation of HFNC is uncertain. The patients in our study had baseline ROX index averages > 5 and did not subsequently fall below baseline. However, after initiation of F_I_O_2_ ≥ 0.4, 26% of the patients were intubated, and most of them (87%) were in the non-prone group. Avoidance of tracheal intubation is reportedly effective in reducing harm to the lungs in terms of lung damage [22,23]. The prolonged length of mechanical ventilation is a known cause of ventilator-related lung injury that positive pressure ventilation creates. Additionally, in the prone position, there is less hyperinflation in the non-dependent lung regions and less cyclic opening and closing of the dependent airspace in ventilated patients [24,25], and some mechanisms may be similar under spontaneous breathing [26]. Thus, the introduction of prone therapy was protective to the lungs in that it resulted in antitussive cough and improved dyspnea in some cases. Coughing generates high intrathoracic pressure [27], which may be harmful to damaged lungs, there are reports of pneumothorax and mediastinal emphysema even with spontaneous breathing in patients with COVID-19 [28,29,30], and it is easy to imagine the possibility of pressure trauma associated with tissue fragility. Therefore, prone positioning may be effective in reducing the use of high-concentration oxygen, rate of tracheal intubation, antitussive cough, and maintaining homogeneous lungs, reducing harm to the lungs, and may lead to better clinical outcomes.

Meanwhile, in the five cases where prone therapy was attempted but daily sessions were abandoned, more than half of the patients had tachypnea (80%) and dyspnea (60%) at the time of prone positioning. Thus, if tachypnea and dyspnea are present at the time of prone positioning, prone positioning therapy should be attempted after improving these symptoms. In some cases, there may be no other means of reducing tachypnea (respiratory workload) other than tracheal intubation and positive pressure ventilation. In contrast, ten (37%) patients who were introduced to the prone position reported improvement in dyspnea, so attempting the prone position may be better before deciding on tracheal intubation.

Our findings have several implications for clinical settings. In the non-invasive management of patients with COVID-19, the interests of treatment and the safety of the healthcare provider must be considered [31]. In many cases, once the patient is in prone and settled, the healthcare provider can be away from the patient for a period of time without the need for contact, although individual evaluation is necessary. We believe that the active introduction of oxygenation is desirable in terms of the benefits to both medical personnel and patients when oxygenation is stable. In Japan, it is difficult to track the effects of prone positioning therapy in detail because most COVID-19 treatment systems move from one facility to another depending on the severity of the disease and the time of year. In our institution, although the management wards differed according to the severity of the disease, including the ICU, detailed follow-up was possible because the patients were managed in the same institution for up to 60 days. Therefore, our data will be important for investigating the effects of prone therapy in Japan.

This study had some limitations. First, because it was a before–after historical cohort study, there may be differences in treatment strategies, especially in the timing and considerations for the indication of tracheal intubation at different times. Although we confirmed that there were no significant differences in serum ferritin, D-dimer, lactate dehydrogenase, and CRP levels, which are markers of COVID-19 severity [32,33], between the two groups, there may be a potential selection bias. Additionally, the differences in the use of Remdesivir between the groups also may have influences the clinical course; therefore, we cannot argue against this effect.

Second, since this was a harmless retrospective observational study without any benefits associated with participation, we aimed to recruit as many participants as possible without a predefined sample size. Therefore, we were not able to investigate the patients’ detailed respiratory conditions, other than their medical history. Moreover, this study used SpO_2_/F_I_O_2_ trends as an assessment of oxygenation, mainly because arterial blood gas analysis was not performed except at the time of admission; thus, partial pressure of arterial oxygen (PaO_2_) could not be measured throughout the course of the study. Therefore, it was not possible to evaluate oxygenation, strictly reflecting the PaO_2_.

Third, the prone position protocol in this study is unique. Therefore, this study might be impossible to compare its effectiveness with the previous research strictly. Nevertheless, its endpoints of prone therapy and outcomes are similar to the previous study [14] and do not differ from the consensus and guidelines for acute respiratory management goals. In addition, since the endpoint of the prone position has not been thoroughly discussed previously, the prone position was performed until discharge.

## 5. Conclusions

Prone treatment of non-intubated patients with COVID-19 respiratory failure who required F_I_O_2_ ≥ 0.4 was associated with higher SpO_2_/F_I_O_2_ at 1 week and 2 weeks after the initiation of F_I_O_2_ ≥ 0.4. Additionally, the clinical endpoints of patients with prone treatment, assessed using the seven-category ordinal scale, showed a tendency to recover along with a time trend. It is a harmless and valuable treatment for the lungs because it can be managed with relatively low oxygen levels, avoids intubation, and reduces excessive dry cough. Awake prone positioning may contribute to the improvement of oxygenation if the therapy initiates by the time at which, at least, oxygen therapy required F_I_O_2_ ≥ 0.4.

## Figures and Tables

**Figure 1 healthcare-10-01006-f001:**
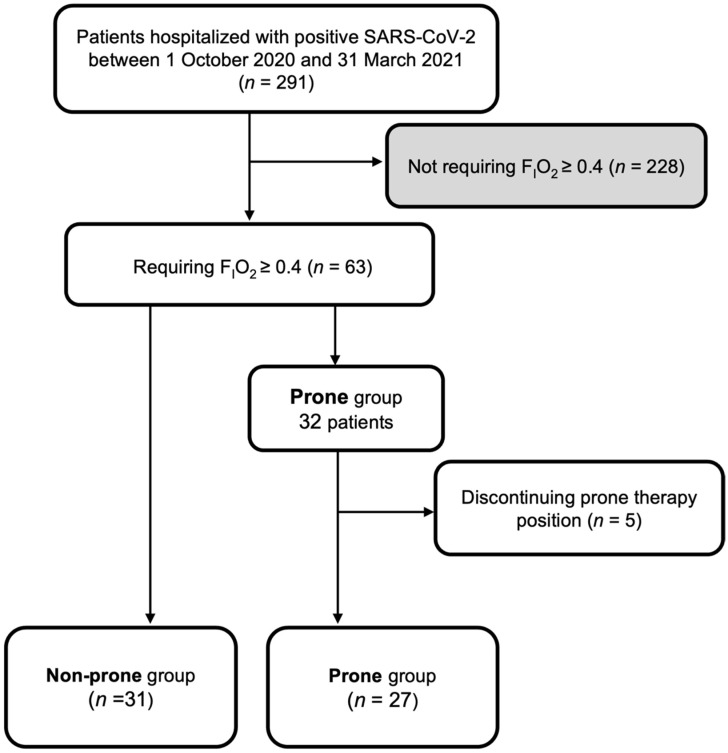
Patient flowchart. F_I_O_2_, fraction of inspired oxygen.

**Figure 2 healthcare-10-01006-f002:**
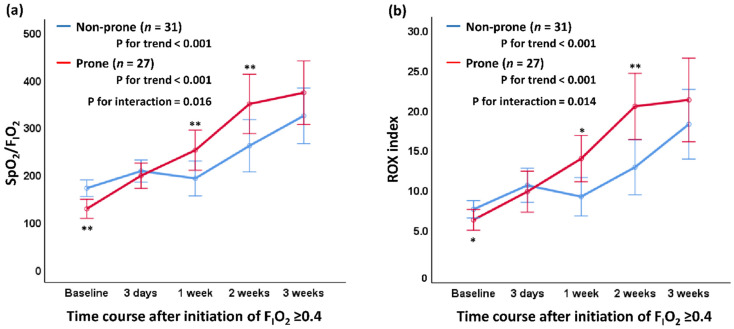
SpO_2_/F_I_O_2_ and ROX index after initiation of F_I_O_2_ ≥ 0.4 between the two groups based on prone therapy. (**a**) changes in SpO2/FIO2 and (**b**) changes in ROX index; red line, prone group; blue line, non-prone group. Data are estimated mean and 95% confidence interval. F_I_O_2_, fraction of inspired oxygen. *, *p* < 0.05 of inter-group comparison. **, *p* < 0.01 of inter-group comparison.

**Figure 3 healthcare-10-01006-f003:**
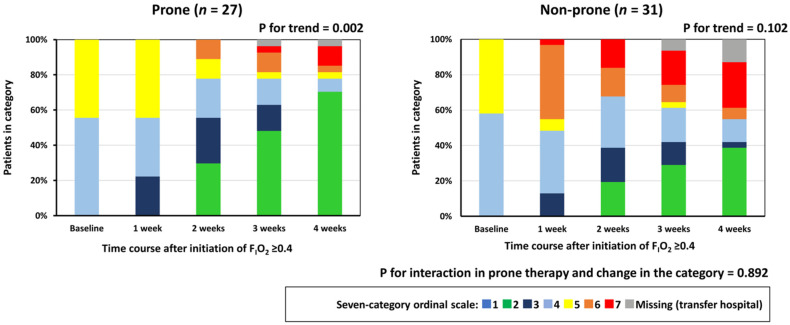
Seven-category ordinal scale after initiation of F_I_O_2_ ≥ 0.4 between the two groups based on prone therapy. The seven-category ordinal scale consisted of the following categories: 1, not hospitalized with resumption of normal activities; 2, not hospitalized, but unable to resume normal activities; 3, hospitalized, not requiring supplemental oxygen; 4, hospitalized, requiring supplemental oxygen; 5, hospitalized, requiring nasal high-flow oxygen therapy, non-invasive mechanical ventilation, or both; 6, hospitalized, requiring ECMO, invasive mechanical ventilation, or both; and 7, death. Data indicate proportion of patients in each category. F_I_O_2_, fraction of inspired oxygen.

**Table 1 healthcare-10-01006-t001:** Baseline patient characteristics between the two groups based on prone therapy.

	All Patients *n* = 58	Non-Prone *n* = 31	Prone *n* = 27	*p* Value
Age, years,	67 (49–75)	63 (49–70)	71 (55–77)	0.75
Female, *n* (%)	15 (22)	8 (13)	7 (10)	0.61
BMI, kg/m^2^	25 (23–30)	24 (23–30)	25 (23–30)	0.13
Medical history *n (%)*				
Hypertension	25 (43)	12 (39)	13 (48)	0.32
Diabetes	26 (45)	15 (48)	11 (41)	0.37
Hyperlipidemia	9 (16)	6 (19)	3 (11)	0.31
Chronic kidney disease	5 (9)	3 (10)	2 (7)	0.56
Hemodialysis	4 (7)	3 (10)	1 (4)	0.36
COPD	3 (5)	2 (6)	1 (4)	0.55
Asthma	5 (9)	1 (3)	4 (15)	0.13
Interstitial pneumonia	1 (2)	0	1 (4)	0.46
Current smoker	4 (7)	1 (3)	3 (11)	0.25
Blood examination on admission				
Serum ferritin, ng/dL	549 (309–1238)	613 (302–1485)	495 (303–905)	0.95
FDP, μg/mL	4.6 (3.8–6.1)	4.6 (2.7–8.7)	4.6 (4.3–5.9)	0.60
D-dimer, μg/mL	1.33 (1.01–1.92)	1.38 (0.94–2.46)	1.27 (1.09–1.91)	0.26
LD, U/L	352 (285–448)	352 (276–450)	337 (287–446)	0.25
CRP, mg/dL	5.3 (3.0–18.1)	5.0 (3.1–19.5)	5.6 (2.9–12.9)	0.88

Data, median (interquartile); BMI, body mass index; COPD, chronic obstructive pulmonary disease; CRP, C-reactive protein; FDP, fibrinogen-fibrin degradation product; F_I_O_2_, fraction of inspired oxygen; LD, lactic acid dehydrogenase; PaO_2_, partial pressure of arterial oxygen.

**Table 2 healthcare-10-01006-t002:** Treatment information and respiratory condition.

	All Patients *n* = 58	Non-Prone *n* = 31	Prone *n* = 27	*p* Value
Do-not-intubate order	4	3	1	0.36
Time between symptom onset and inhalation > F_I_O_2_ ≥ 0.4, days	9 (7–10)	9 (7–10)	10 (7–11)	0.74
PaO_2_/ F_I_O_2_ at initiation F_I_O_2_ ≥ 0.4	120 (81–191)	166 (89–260)	117 (70–150)	0.10
Respiratory rate at initiation F_I_O_2_ ≥ 0.4, /min	23 (22–27)	26 (22–28)	23 (20–25)	0.10
Respiratory rate at initiation F_I_O_2_ ≥ 0.4, /min	22 (20–26)	24 (22–26)	22 (20–23)	0.869
Respiratory rate at 3 days after initiation F_I_O_2_ ≥ 0.4, /min	20 (18–23)	22 (17–25)	21 (20–23)	0.807
Respiratory rate at 1 week after initiation F_I_O_2_ ≥ 0.4, /min	19 (18–22)	22 (20–23)	20 (18–22)	0.622
Respiratory rate at 2 weeks after initiation F_I_O_2_ ≥ 0.4, /min	19 (18–22)	21 (18–23)	19 (18–20)	0.105
Respiratory rate at 3 weeks after initiation F_I_O_2_ ≥ 0.4, /min	18 (16–22)	18 (16–22)	19 (18–21)	0.730
F_I_O_2_ at initiation F_I_O_2_ ≥ 0.4	0.60 (0.40–0.70)	0.50 (0.40–0.60	0.60 (0.60–0.75)	0.039
F_I_O_2_ at 3 days after initiation F_I_O_2_ ≥ 0.4	0.40 (0.39–0.60)	0.40 (0.40–0.50)	0.40 (0.32–0.60)	0.311
F_I_O_2_ at 1 week after initiation F_I_O_2_ ≥ 0.4	0.40 (0.24–0.60)	0.40 (0.28–0.53)	0.30 (0.21–0.55)	0.076
F_I_O_2_ at 2 weeks after initiation F_I_O_2_ ≥ 0.4	0.24 (0.21–0.36)	0.26 (0.21–0.39)	0.21 (0.21–0.24)	0.219
F_I_O_2_ at 3 weeks after initiation F_I_O_2_ ≥ 0.4	0.21 (0.21–0.28)	0.21 (0.21–0.32)	0.21 (0.21–0.25)	0.537
Oxygen delivery interface initiation of F_I_O_2_ ≥ 0.4, *n* (%)				
Simple oxygen mask	32 (55)	17 (55)	15 (56)	1.000
High-flow nasal canula	25 (43)	14 (45)	11 (41)	0.795
NPPV	1 (2)	0	1 (3)	0.466
Symptom, *n* (%)				
Dry cough	24 (41)	10 (32)	14 (52)	0.18
Fever up (> 38.0 °C)	9 (16)	7 (23)	2 (7)	0.15
Confusion (Conscious disturbance)	7 (12)	3 (10)	4 (15)	0.69
Tachypnea (respiratory rate >25/min)	19 (33)	12 (39)	7 (26)	0.40
Dyspnea	29 (50)	19 (61)	10 (37)	0.06
Medical treatment, *n* (%)				
Favipiravir	22 (38)	13 (42)	9 (33)	0.59
Remdesivir	20 (34)	6 (19)	14 (52)	0.01
Steroid	55 (95)	29 (93)	26 (96)	0.99
Rescue therapy/palliative care, *n* (%)				
Fentanyl	4 (14)	2 (6)	2 (7)	1.000
Morphine	1 (1)	1 (3)	0	1.000
Dexmedetomidine	5 (8)	3 (9)	2 (7)	1.000

**Table 3 healthcare-10-01006-t003:** Clinical outcomes according to prone positioning.

*Outcomes*	All *n* = 58	Non-Prone *n* = 31	Prone *n* = 27	*p* Value
Tracheal intubation, n (%)	15 (26)	13 (42)	2 (7)	0.003
Outcome of hospitalization, n (%)				
Discharge without supplemental oxygen	31 (53)	13 (42)	18 (67)	0.084
Discharge requiring supplemental oxygen	5 (9)	3 (10)	2 (7)	0.603
Transfer hospital requiring supplemental oxygen	5 (9)	3 (10)	2 (7)	0.603
Transfer hospital without supplemental oxygen	5 (9)	3 (10)	2 (7)	0.603
Death	11 (19)	8 (26)	3 (11)	0.170
In the general ward of our hospital	2 (3)	0	2 (7)	0.212
Day of hospitalization	20 (15–31)	24 (15–31)	20 (15–25)	0.934

**Table 4 healthcare-10-01006-t004:** Details of patients who could not sustain prone positioning.

	*n* = 5
Symptoms, *n* (%)	
Dry cough	1 (20)
Fever up (>38.0 °C)	2 (40)
Confusion (Conscious disturbance)	0
Tachypnea (RR > 25)	4 (80)
Dyspnea	3 (60)
Outcomes, *n* (%)	
Discharge without supplemental oxygen	2 (40)
Discharge requiring supplemental oxygen	0
Transfer hospital requiring supplemental oxygen	0
Transfer hospital without supplemental oxygen	1 (20)
Death	2 (40)

## Data Availability

Not applicable.

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
