# Peer review of "Detailed Changes in Oxygenation following Awake Prone Positioning for Non-Intubated Patients with COVID-19 and Hypoxemic Respiratory Failure—A Historical Cohort Study"

_healthcare, 2022, doi:10.3390/healthcare10061006_

Round 1
Reviewer 1 Report
The manuscript entitled “Derailed changes in oxygenation following awake prone positioning for non-intubated patients with COVID-19 and hypoxemia respiratory failure-a historical cohort study” was reviewed. The authors performed a retrospective study and review of 58 patients with COVID-19 who were admitted to the hospital, from 2020, 10.1 to 2021. 3. 31, and a cohort review for 3 weeks. The authors found that SPO2/FiO2, ROX index, and the seven-category ordinal scale were significantly different in the prone group, may improve oxygenation within two weeks, and some detailed data. I have some comments for this study:
- Currently, there were many studies revealing the effects of prone positioning in awake patients with COVID-19. So, your prone therapy protocol needs to be enhanced in the paper, such as start timing? of 58 patients were on FiO2 40%? What is the endpoint? etc.
What was the main recommendation from the study results? early use the awake prone or when FIO2 40% start to use it or later use it after PaO2/FiO2 lower still to 9 days if FiO2 40%?
- In figure 1, the authors may note reasons in non-prone therapy if all admitted patients were required the awake prone as FiO2 40%. All patients tried 30 minutes of the prone positioning process.
- In table 1, I found the COVID patient was 9 days on average days of time between symptom onset and FiO2 >=40%, but PaO2/FiO2 at admission was 120, which is too low in clinical concept for these days? So, before FiO2 is 40% what concept do operate FiO2 or maintain PaO2/FiO2 level in the patient with hypoxemia?
- In Table 1, the face mask may be revised, in line 132 of the manuscript, all the O2 therapy devices were used in the study? what are the types of O2 therapy devices on the facemask? And nasal cannula?
- In Table 1 the exhibition is too long, and not easy to read.
- In Table 3, FIO2 pre prone positioning, what is the meaning? On the day of start prone? FiO2 40%? FIO2 3 days after starting prone positioning, what is the meaning? Indicating on the third day?
- Why was detail analysis only in the prone group? the non-prone group?
- In line 282, we set certain standards for the timing of intubation. What is it, should be described in the method?
- In Table 1, PaO2/FiO2 at admission was not significantly different in both groups, in figure 2, SPO2/FiO2 on the baseline was significantly different in inter-group, so the baseline is when?
- In figure 2, the trend of SPO2/FiO2 was also significantly increased in the non-prone group, what should the oxygenation target be in COVID patients in the hospital in the study? In the discussion paragraph, may discuss the prone positioning protocol in other studies of COVID19 patients.
Reviewer 2 Report
#1
The word "inadequate" in the last sentence of the first paragraph of the introduction is not appropriate.
#2
I believe that not all patients in the prone group continued prone therapy until the end of the observation period.
How was the number of days of prone therapy in this study determined?
The protocol for prone therapy should include a statement regarding "criteria for discontinuation of prone therapy".
#3
The number of days of prone therapy should be described as to how it was included in the analysis.
#4
If the patient's respiratory status deteriorated again after prone therapy was terminated, please describe how the patient was handled in analysis.
#5
There is a significant difference in the use of Remdesivir between the Prone and Non-prone groups. This may have influenced the improvement in disease status. This needs to be discussed in the manuscript.
Round 2
Reviewer 1 Report
I have no question again.
Reviewer 2 Report
I believe this manuscript has been appropriately revised.